# Responses in Plant Growth and Root Exudates of *Pistia stratiotes* under Zn and Cu Stress

**DOI:** 10.3390/plants13050736

**Published:** 2024-03-05

**Authors:** Yujie Wang, Pan Zhang, Canhua Yang, Yibai Guo, Panpan Gao, Tong Wang, Yu Liu, Lina Xu, Gongke Zhou

**Affiliations:** 1College of Landscape Architecture and Forestry, Qingdao Agricultural University, Qingdao 266109, China; yuji_w@163.com (Y.W.); hh18754198168@163.com (P.Z.); yangcanhua0330@163.com (C.Y.); 13156026005@163.com (Y.G.); 15763793751@163.com (P.G.); ecotone@qau.edu.cn (T.W.); zhougk@qau.edu.cn (G.Z.); 2Shandong Key Laboratory of Eco-Environmental Science for the Yellow River Delta, Binzhou University, Binzhou 256600, China

**Keywords:** heavy metal, aquatic plant, plant growth, root exudate, aquatic environment

## Abstract

At present, the situation regarding heavy metal pollution in aquatic environments is becoming more and more serious. The bioaccumulation of heavy metals in aquatic plants causes obvious phytotoxicity, which can also induce secondary pollution in the aquatic environment. Zinc and copper, as indispensable elements for plant growth, are also prominent heavy metals in water pollution in China, and their concentrations play a crucial role in plant growth. In this study, we investigated the response of *Pistia stratiotes* (*P. stratiotes*) to different concentrations of Zn and Cu, and the results showed that plant growth and photosynthesis were inhibited under both Zn (1, 2, 4, and 8 mg/L) and Cu (0.2, 0.4, 0.8, and 1 mg/L) stresses. The relative growth rates of *P. stratiotes* under 8 mg/L Zn or 1 mg/L Cu stress were 6.33% and 6.90%, which were much lower than those in the control group (10.86%). Meanwhile, Zn and Cu stress caused insignificant change in the relative water contents of plants. The decrease in phlorophyll fluorescence parameters and chlorophyll contents suggested the significant photoinhibition of Zn and Cu stress. Chemical analysis of plant root exudates showed that the root secretion species obtained by gas chromatography–mass spectrometry (GC–MS) mainly included amino acids, alkanes, aldehydes, ketones, phenols, and more. Compared with the control group, the influence of Zn or Cu on the reduction in relative amounts of exudates was greater than that on the increase. The results of this study provide important data for the utilization of *P. stratiotes* in heavy metal-polluted water environments.

## 1. Introduction

With the rapid development of urbanization, water quality has been seriously affected by the increasing activities of the population, human activities, fast industrialization, unskilled utilization of natural water resources, and so on [1,2]. Heavy metal pollution in the aquatic environment is a critical concern due to the potential toxicity of heavy metals and their accumulation in aquatic habitats [3,4]. Heavy metals are a group of metals and metalloids that have an atomic density of more than 4000 kg/m^3^ [5]. Heavy metals such as copper (Cu), cadmium (Cd), zinc (Zn), chromium (Cr), arsenic (As), boron (B), cobalt (Co), titanium (Ti), strontium (Sr), tin (Sn), vanadium (V), nickel (Ni), molybdenum (Mo), mercury (Hg), and lead (Pb) are commonly found in global river and lake water [6,7]. It is reported that the mean heavy metal concentrations in surface water are greater than the published threshold limits, as per the standards of both the World Health Organization (WHO) and the United States Environmental Protection Agency (USEPA), in the developing countries of Africa, Asia, and South America [8]. In China, the detected concentrations of Cu and Zn in the Wen–Rui Tang River were 96.8 μg/L and 98.3 μg/L, respectively [9]. Guo et al. reported that 11 heavy metals were detected in the South-to-North Water Diversion Project, and Cu (0.02–185.4 μg/L) and Zn (0.5–134.9 μg/L) had higher concentrations than other heavy metals [10]. Zhang et al. reported that only Zn and Cu were found in the water of Tangbai River in China, and their contents were 0.59 ± 0.25 and 0.37 ± 0.09 mg/L, respectively [11]. Meanwhile, Zn and Cu are elements vital for plant growth. However, when exposed to greater amounts of these metals, plants are harmed.

It is well known that the response of plant growth to stress induced by heavy metals, such as Zn and Cu, is concentration- and species-specific [12]. To date, adverse effects have been reported after using Zn and Cu on plants, although many of these plant toxicity studies have focused on terrestrial plants. It is reported that Zn (50 μM) can significantly reduce the shoot dry weight, total plant biomass, and total chlorophyll content of *Solanum lycopersicum*, as confirmed through hydroponic experiments [13]. Paunov et al. found that Zn (600 μM) decreased the relative growth rate, photosynthesis, and the inactivation of photosystem II reaction centers in *Triticum durum* [14]. In addition, reductions in plant biomass and the inhibition of root growth are the common symptoms associated with excess heavy metals. Concentrations of Cu at 0.2, 5.0, 25, and 50 μM all impair the growth of *Lolium perenne*, with a decrease in root–shoot biomass, leaf size and area [15]. However, data regarding the influence of Zn and Cu at different concentrations on aquatic plants are still lacking in relevant research. In the aquatic environment, aquatic plants can remediate heavy metal pollution though absorption [16], while the bioaccumulation of heavy metals in the plants causes obvious phytotoxicity, which can also induce secondary pollution in the aquatic environment. Therefore, it is necessary to study the concentration-dependent effect of Zn and Cu on the phytotoxicity of aquatic plants. Importantly, aquatic plants also release organic matter during their growth and metabolism, which accounts for 20–40% of the photosynthetically fixed carbon, and it is then transferred to the rhizosphere through root exudates [17]. Root exudates can influence the mobility and phytoavailability of heavy metals [18], and also provide bounteous energy and carbon for microbes and plankton in the aquatic environment [19]. Hence, the negative effects of heavy metals on the root exudates of aquatic plants can pose risk to the aquatic environment.

*Pistia stratiotes* (*P. stratiotes*) is a typical floating plant, with rapid growth and a well-developed root system which can come into direct contact with pollution. *P. stratiotes* is tolerant to various environmental conditions and can accumulate or absorb the contaminants from the water through its different parts; it is regarded as a low-cost and solar-energy-driven source of phytoremediation [20]. It is reported that *P. stratiotes* presents different accumulation and tolerance levels under different metal (Ag, Cd, Cr, Cu, Hg, Ni, Pb, and Zn) conditions [21]. The removal rates of Cr and Cu via the absorption of *P. stratiotes* are 77.3% and 80.9%, respectively [22]. Therefore, *P. stratiotes* is particularly interesting for the researchers dealing with water environmental pollutants. However, the understanding of the toxicity of heavy metals Zn and Cu on *P. stratiotes* is not comprehensive. Hence, *P. stratiotes* is used in this work to investigate (1) the growth inhibition of Zn and Cu; (2) the response of the photosystem after Zn and Cu exposure; (3) and the change in root exudates induced by Zn and Cu. These findings will help us to gain insight into the negative effect of Zn and Cu on the growth and root exudates of *P. stratiotes* and will provide critical data to inform the use of *P. stratiotes* on heavy metal pollution in aqueous environments.

## 2. Results and Discussion

### 2.1. Effects on Growth

As shown in Appendix A, after exposure to low concentrations of Zn (0.2, 0.4, 0.6, and 0.8 mg/L) for 20 days, the fresh weight (FW) and dry weight (DW) of the shoot and root were insignificantly changed, while the biomass of the shoot and root exhibited a decreasing tendency upon 1 mg/L Zn exposure. Therefore, we chose the higher concentration of Zn (1, 2, 4, and 8 mg/L) for further experiments. The growth of *P. stratiotes* after Zn (1, 2, 4, and 8 mg/L) or Cu (0.2, 0.4, 0.8, and 1 mg/L) exposure was detected and is shown in Figure 1. The FW and DW of *P. stratiotes* continuously increased with the extension of treatment time (0, 5, 10, 15, 20 day). For the control group, the FW of *P. stratiotes* increased by 69.98% after growing for 20 days, while both Zn- and Cu-treated plants had significantly lower FW compared with that in the control group (Figure 1A,B). The FW of *P. stratiotes* increased by 55.10%, 33.81%, 25.49% and 9.80% after growing at Zn concentrations of 1, 2, 4 and 8 mg/L for 20 days, respectively, as opposed to the control group. Similar trends were observed in plants grown at Cu concentrations of 0.2, 0.4, 0.8, and 1 mg/L, with the increase in FW of 52.87%, 46.52%, 32.05%, and 16.02%, respectively (Figure 1B). In addition, the effects of Zn and Cu on the DW had the same trend as those on the FW (Figure 1C,D). After being treated with Zn at concentrations of 0, 1, 2, 4, and 8 mg/L for 5 days, the DWs of *P. stratiotes* were measured as 1.12, 1.09, 1.06, 1, and 0.96 g, respectively, which were insignificantly changes among the different treated concentrations of Zn. However, with the increase in the treatment time, the DW of *P. stratiotes* exhibited the greatest change among the different treatment concentrations of Zn. The same trend was observed in the case of Cu treatment. After exposure to the highest concentration of Zn or Cu for 20 days, the relative water contents of plants gradually decreased, but the variation was insignificant and the relative water contents remained essentially between 91% and 93% (Figure 2A). It was suggested that Zn and Cu stress caused insignificant changes in the relative water contents of plants. In addition, the relative growth rates of *P. stratiotes* were calculated and are shown in Figure 2B. The relative growth rates of *P. stratiotes* under 8 mg/L Zn and 1 mg/L Cu stress were 6.33% and 6.90%, which were much lower than that in control group (10.86%). These results further proved the phytotoxicity of Zn (1–8 mg/L) and Cu (0.2–1 mg/L) towards *P. stratiotes*. Additionally, the growth inhibition of *P. stratiotes* in Cu stress was greater than that of Zn stress, which was supported by the earlier literature [12].

### 2.2. Effect on Chlorophyll Fluorescence Parameters and Chlorophyll Contents

Chlorophyll fluorescence parameters were found to be strongly correlated with whole-plant growth following environmental stress and therefore can be considered reliable indicators of the intensity of stress [23]. Among these parameters, Fv/Fm is the maximum photochemical quantum yield of PS II or the light energy conversion efficiency of maximum PS II. It is commonly used as a crucial indicator of the photosynthetic system, measuring the PSII primary light energy conversion efficiency of leaves and indicating the response of plants to stress [24]. As shown in Figure 3A,B, the Fv/Fm values of the control group showed stable changes, fluctuating between 0.78 and 0.80. Hence, the decreases in Fv/Fm after exposure to Zn and Cu were concentration- and time-dependent. On the 20th day, Zn exposure at concentrations of 1, 2, 4, and 8 mg/L resulted in decreased Fv/Fm values of 0.75, 0.69, 0.64, and 0.57, respectively. They decreased by 3.85%, 11.54%, 17.95%, and 26.92%, respectively, when compared with the blank control. The treatment of Cu at concentrations of 0.2, 0.4, 0.8, and 1 mg/L also resulted in the inhibition of the Fv/Fm values. After 20 days of exposure of Cu, the Fv/Fm values were 0.74, 0.73, 0.72, and 0.71, respectively, which were decreased by 5.13%, 6.41%, 7.69%, and 8.97%, respectively, when compared with the blank control. The results were similar to those of Hayat et al., who concluded that the Fv/Fm of plants under heavy metal stress was lower than that of blank control, and that the higher concentration, the faster and greater the decrease [25]. This indicated that both Zn and Cu stress could reduce the light energy capture efficiency and irreversibly affect the photosynthetic reaction center. Some heavy metals, such as Hg^2+^, Cu^2+^, Cd^2+^, Zn^2+^, and Ni^2+^, replace the central Mg^2+^ atom in the chlorophyll molecule, which lowers the fluorescence quantum yield and shifts the fluorescence spectrum. In addition, the decrease in Fv/Fm value indicated the photoinhibitory or photo-oxidative effects on PS II [26]. Photosynthesis inhibition and the oxidative stress of heavy metals on plants are commonly reported [27,28].

PI_ABS_ is a performance index based on absorbed light energy, which can accurately reflect the overall state of the plant photosynthetic apparatus [29]. As can be seen in Figure 3C,D, the values of PI_ABS_ in the control group were in a steady state. However, Zn (1, 2, 4, and 8 mg/L) exposure significantly inhibited PI_ABS_. After exposure to Zn (1, 2, 4, and 8 mg/L) for 5 days, the decreased rates of PI_ABS_ were 0.12%, 11.93%, 23.87%, and 26.78%, respectively. After 20 days of treatment, the decreased rates were 8.02%, 31.45%, 43.16%, and 49.44%, respectively. In addition, PI_ABS_ values were decreased with the increase in Cu concentrations and treatment times. Upon 5 days of exposure, Cu (0.2, 0.4, 0.8, and 1 mg/L) decreased PI_ABS_ values by 11.76%, 12.22%, 30.65%, and 36.01%, respectively, when compared with the blank control, while PI_ABS_ values decreased by 37.01%, 45.48%, 47.71% and 49.43% at 20 days of exposure. PI_ABS_ could reflect the effects on PS II. It is speculated that PS II electron transfer activity is affected by Zn and Cu exposure [30].

Chlorophyll content plays an important role in plant photosynthesis. The decrease in chlorophyll content in plant leaves under adverse stress directly affects plant photosynthesis, inhibiting plant growth. The chlorophyll contents of leaves after Zn and Cu exposure were shown in Figure 4. In both Zn and Cu treatments, the pigment contents followed the trend: chlorophyll a > chlorophyll b > carotenoids. Chlorophyll a content was found to be more sensitive to stress, showing the highest decrease compared to chlorophyll b and carotenoids. Similar results have also been reported by Sheetal et al. (2016) [31]. Chlorophyll b is a light-collecting pigment, while chlorophyll a is a major antenna pigment that plays a key role in primary photochemical reactions. It has been reported that Cu inhibits a synthesis of d−aminolevulinic acid and protochlorophyll reductase, leading to a decrease in chlorophyll a content [32]. Both Zn and Cu treatment significantly decreased the chlorophyll a content, which suggested the photoinhibition of Zn and Cu exposure. Cui et al. found that Zn reduces the synthesis of chlorophyll and adversely affects the growth of *Aglaonema* spp. and *Spathiphyllum kochii* through hydroponic experiments, which is consistent with the results of this study [33]. The chlorophyll contents were significantly decreased upon Zn and Cu exposure. However, the different concentrations of Zn did not induce significant changes in the contents of chlorophyll a, chlorophyll b, and carotenoids. Concentration-independent photoinhibition of Zn is suggested. On the other hand, Cu exhibited a greater inhibition of chlorophyll content compared to Zn, showing a concentration-dependent effect. The decrease in chlorophyll contents can be attributed to the direct inhibition of enzymes or the competitive exclusion of some essential nutrients [31].

### 2.3. Root Exudates Analysis

Root exudates mainly consist of carbon-containing compounds, including sugars, amino acids, fatty acids, organic acids, peptides, and so on [34]. The composition and relative amount of root exudates under heavy metal stress can reflect subtle changes in the water micro-environment [35]. The compositions and the relative contents of root exudates in *P. stratiotes* after different treatments were tabulated in Appendix A. As shown in Appendix A, a total of 48 compounds were detected in normal *P. stratiotes*, including amino acids, alkanes, aldehydes, ketones, phenols, and more. However, after exposure to Zn (8 mg/L) or Cu (1 mg/L), the root exudates were found to contain 56 and 59 compounds, respectively, indicating a greater number of compounds compared to normal plants. According to Venn diagram analysis (Figure 5), only one hydrocarbon (Bicyclo[4.4.1]–undeca–1,3,5,7,9–pentaene) was detected in both the root exudates of the control plant and the Zn-treated plant. 1–Heptatriacotanol, E,E,Z–1,3,12–Nonadecatriene–5,14–diol, and tetrapentacontane, 1,54–dibromo– were the shared exudates in the control and Cu-treated plant. 1–Heptatriacotanol is involved in plant immune regulation, which can improve self-protection against adversity stress [36]. One report suggested that hydrocarbon tetrapentacontane could positively and negatively regulate the rhizosphere microbial diversity [37]. Khromykh et al. reported that tetrapentacontane is correlated with metabolic reprogramming in the epidermal cells of infected leaves of *Prunus species* [38]. Upon Zn and Cu treatment, 13 compounds were detected in the exudates. In addition, 25 compounds were found, including alkanes, amino acids, and thiols. The relative contents of the compounds of exudates were analyzed and the results are shown in Figure 6 and Appendix A. Compared with the normal plants, the relative contents of 13 compounds upon Zn treatment were increased, and those of 12 compounds were inhibited. Therein, the relative content of 7,9–Di–tert–butyl–1–oxaspiro(4,5)deca–6,9–diene–2,8–dione increased by 3.35%, and this compound was identified as a repeatedly present contaminant in the migration waters [39]. Furthermore, the contents of other compounds increased by less than 1%. The decreased relative contents of phenol,2,2′–methylenebis[6–(1,1–dimethylethyl)–4–methyl–, pentadecane, hexadecane,2,6,11,15–tetramethyl–, pentacosane, and heneicosane were 6.1%, 3.35%, 2.85%, 1.7%, and 1.05%, respectively. Phenol,2,2′–methylenebis[6–(1,1–dimethylethyl)–4–methyl– is a typical phenolic antioxidant, which is used as a potential active compound against protein tyrosine phosphatase 1B [40]. Hexadecane–tetramethyl is used as a hydrocarbon biomarker for archaea identification in geographical situations [41]. These results suggested that the oxidation resistance of root exudates was inhibited, and the water quality was affected under Zn stress. In the case of Cu stress, the relative contents of 8 compounds in the exudates showed a small increase of less than 1%. However, the relative contents of 16 compounds decreased compared with the control group. Therein, the decrease in the contents of hexadecane–tetramethyl–, phenol, 2,2′–methylenebis[6–(1,1–dimethylethyl)–4–methyl–, pentadecane, pentacosane, eicosane, and heptadecane were 6.45%, 6.25%, 3.6%, 1.45%, 1.4%, and 1.1%, respectively. It is reported that pentadecane and eicosane participated in the anti-microbial, anti-inflammatory, proliferation, migration, and collagen synthesis [42,43]. Additionally, among the specific volatile components of root exudates, pentacosane and heptadecane were identified as the specific allelopathic components in root exudates, suggesting their potential role in signal communication [44,45]. These findings indicate the negative effects of Cu and Zn on the plant resistance and the rhizosphere environment. It is worth noting that Cu had the greater toxicity on both plant growth and root exudate composition, highlighting the need for increased attention to Cu pollution in aquatic environments.

## 3. Materials and Methods

### 3.1. Plant Material and Cultivation

*P. stratiotes* plants were obtained from the local marketplace in Qingdao city, China. The plants were carefully washed with deionized water to remove dust and other organic matter, and then placed in the Hoagland nutrient solution for 10 days to acclimatize. The Hoagland solution contained all essential elements at the appropriate levels for the growth of a wide range of plants. Healthy and mature *P. stratiotes* individuals at a similar growing stage (4-leaf) were selected for further experiments. The plant material was analyzed at the beginning of the experiment.

Synthetic metal solutions were prepared by dissolving certain quantities of metals (Zn^2+^ in the form of ZnSO_4_·7H_2_O and Cu^2+^ in the form of CuSO_4_·5H_2_O) in the Hoagland solution. *P. stratiotes* plants were transferred to the different concentrations of solutions and grown for 20 days. The healthy plants cultivated in the Hoagland solution without metal solution exposure were marked as the control group (CK). Each treatment had 6 plants with four replicates. During the treatments, the Hoagland solution was supplemented with the appropriate treatment solutions to maintain the initial volume of 3L.

### 3.2. Biomass

The plants were collected and further analyzed at different treatment times. The plants treated with Zn and Cu were carefully rinsed with deionized water. Filter paper was used to dry the leaves, and then the FW of plants was determined. The DW was determined after drying the plants in an oven at 105 °C for 30 min and then at 80 °C until a constant weight was achieved.

The relative water content (RWC) of the plants indicates the amount of water retained in different fractions by each species as a percentage of dry weight. RWC value was calculated using Equation (1) [46]:RWC (%) = (W_f_ − W_d_)/W_f_ × 100(1)
where W_f_ = fresh weight (g); W_d_ = dry weight after drying the plants in oven (g).

Relative growth rate (RGR) is a critical parameter used in the assessment of the physiological response of plant to toxic chemicals [47]. Present RGR value was calculated by the following Equation (2) [48]:RGR (%) = [(ln W_t_ − ln W_0_)/(t_2_ − t_1_)] × 100(2)
where W_t_ = final plant dry weight at day of harvest (g); W_0_= initial plant dry weight (g); t = planting periods after and before harvest (day)

### 3.3. Leaf Chlorophyll Fluorescence and Chlorophyll Content

Chlorophyll fluorescence parameters were measured using the Pocket PEA+ (Hansatech Instruments, King’s Lynn, UK) connected to PEA plus V1.10 software. Prior to measurements, all sample leaves were adapted to darkness for 30 min. Generally, ChlF is measured in vivo after dark adaptation. Raw fluorescence OJIP transients were transferred to a spreadsheet using the Pocket PEA program, along with minimal fluorescence (Fo), maximal fluorescence (Fm), the performance index based on absorbed light energy (PIabs), and so on. PS II maximum light energy conversion efficiency was calculated as Fv/Fm= (Fm − Fo)/Fm, where Fv is the variable fluorescence calculated as Fv = Fm − Fo.

Chlorophyll content was determined after plants were grown in the different concentrations of Zn (1, 2, 4, and 8 mg/L) or Cu (0.2, 0.4, 0.8, and 1 mg/L) for 20 days. The plants in the Hoagland solution were also assessed. Fresh leaves were thoroughly cut and submerged in 10 mL of 96% ethanol in the dark for 2 days at 4 °C until the residues became almost colorless. The absorbance of the extract solutions was measured at 470, 649, and 665 nm, along with the blank of the solvent (96% ethanol), using a spectrophotometer UH5300 (Hansatech Instruments, King’s Lynn, UK). Chlorophyll contents were calculated according to Porra et al. [49].
Chlorophyll a (mg/L) = 13.95 (A665) − 6.88 (A649)(3)
Chlorophyll b (mg/L) = 24.96 (A649) − 7.32 (A665)
Carotenoid (mg/L) = [1000 (A470) − 2.05 Ca − 114.8 Cb]/248
where A665, A649 and A470 = absorbance values of pigment extracts at wavelengths 665 nm, 649 nm, and 470 nm

### 3.4. Root Exudates

At the end of the 20th day, root exudate was collected from the plant roots by immersing them in deionized water for 8 h. The root exudate samples were filtered through a filter (0.22 μm), and then freeze-dried for 8 h. The samples were subsequently dissolved in 1 mL of n−hexane and analyzed using gas chromatography–mass spectrometry (GC–MS). The following chromatographic conditions were set up according to the method described by Usharani and Vasudevan [50].

### 3.5. Statistical Analysis

All data presented are mean values and standard deviations (SDs). The data were statistically analyzed using one-way analysis of variance in SPSS (version 19.0). One-way ANOVA and least significant difference (LSD) methods were used to test the differences among treatments.

## 4. Conclusions

The results indicated that Zn and Cu stress inhibited plant growth in a concentration- and time-dependent manner. The decrease in Fv/Fm, PI_ABS_, and chlorophyll contents suggested the disruption of chloroplasts, chlorophyll degradation, and impaired electron transfer in the process of plant photosynthesis due to Zn and Cu stress, ultimately leading to a decrease in biomass. Furthermore, the species of root exudates released by *P. stratiotes* were significantly altered under Zn and Cu stress compared to the control group. Root exudation is an effective mechanism employed by plants to cope with and modify their growth environment, promoting the establishment of beneficial microbial communities, which in turn can promote plant growth. The results in this work showed that Cu and Zn had negative effects on plant resistance and rhizosphere environment. Moreover, Cu was found to be more toxic to plant growth and root exudates, which should raise more concerns about Cu pollution in aquatic environments. These findings provide valuable insights for the application of *P. stratiotes* in the remediation of heavy metal-polluted waters. It is worth noting that the concentrations of Zn and Cu used in the laboratory experiment to induce negative effects on *P. stratiotes* were significantly higher than those typically found in natural environments. Therefore, these results contribute to a better understanding of the accumulation and effects of Zn and Cu in *P. stratiotes* under long-term exposure in natural environments.

## Figures and Tables

**Figure 1 plants-13-00736-f001:**
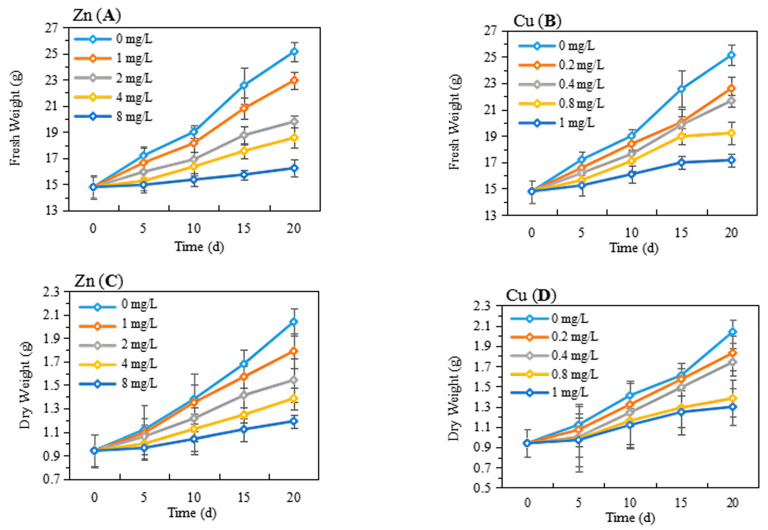
The biomass of *P. stratiotes* after Zn (**A**,**C**) and Cu (**B**,**D**) exposure at the different concentrations and treatment times.

**Figure 2 plants-13-00736-f002:**
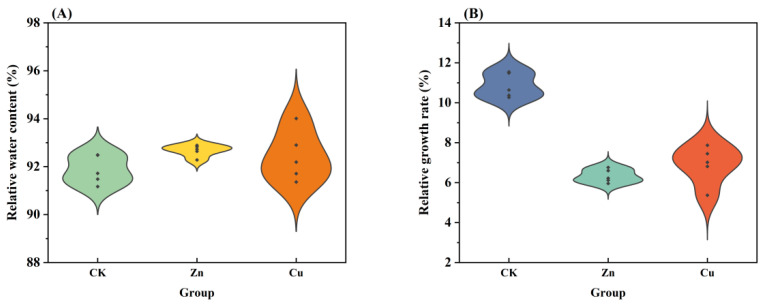
The relative water content (**A**) and relative growth rate (**B**) of *P. stratiotes* in CK (control), Zn (8 mg/L) stress, and Cu (1 mg/L) stress groups at day 20. The parameters of a single sample are shown in each plot (indicated by small black rhombs). CK means the plants cultivated in the Hoagland solution without metal solution exposure.

**Figure 3 plants-13-00736-f003:**
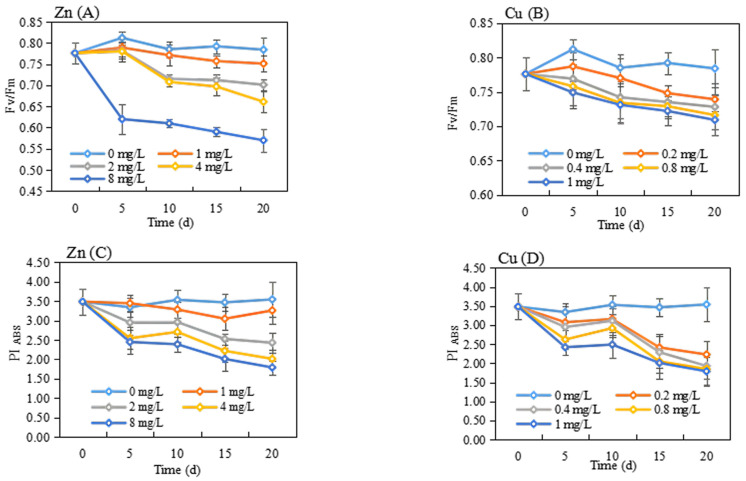
The chlorophyll fluorescence parameters of *P. stratiotes* after Zn (**A**,**C**) and Cu (**B**,**D**) exposure at the different concentrations and treatment times.

**Figure 4 plants-13-00736-f004:**
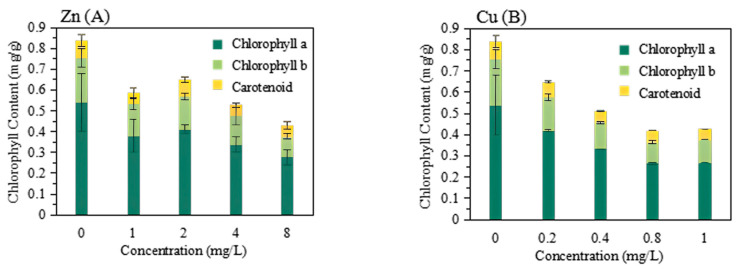
The contents of chlorophyll a, chlorophyll b, and carotenoids of *P. stratiotes* after exposure to Zn (8 mg/L) (**A**) and Cu (1 mg/L) (**B**) for 20 days.

**Figure 5 plants-13-00736-f005:**
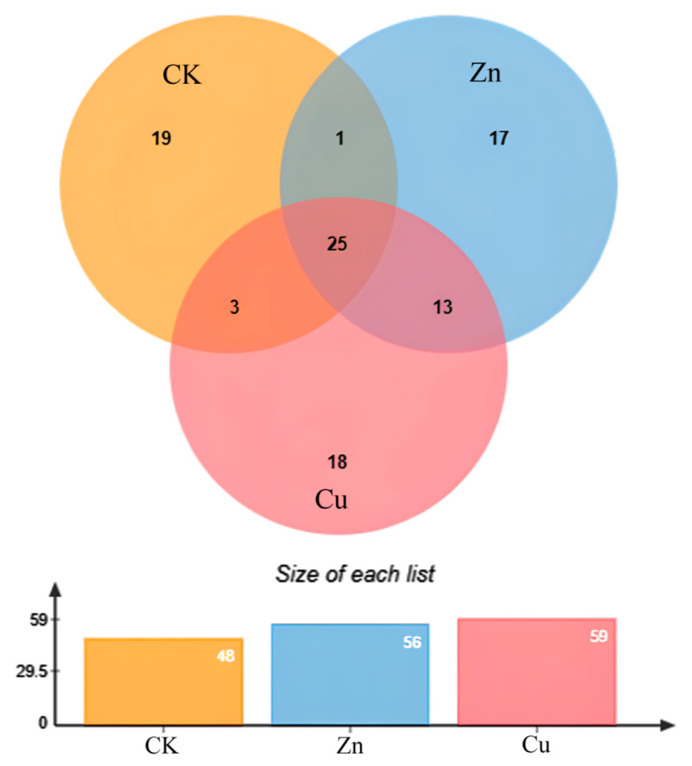
Venn diagram analysis of the compounds of root exudates in CK (control), Zn (8 mg/L) and Cu (1 mg/L) treatment for 20 days. CK means the plants cultivated in the Hoagland solution without metal solution exposure.

**Figure 6 plants-13-00736-f006:**
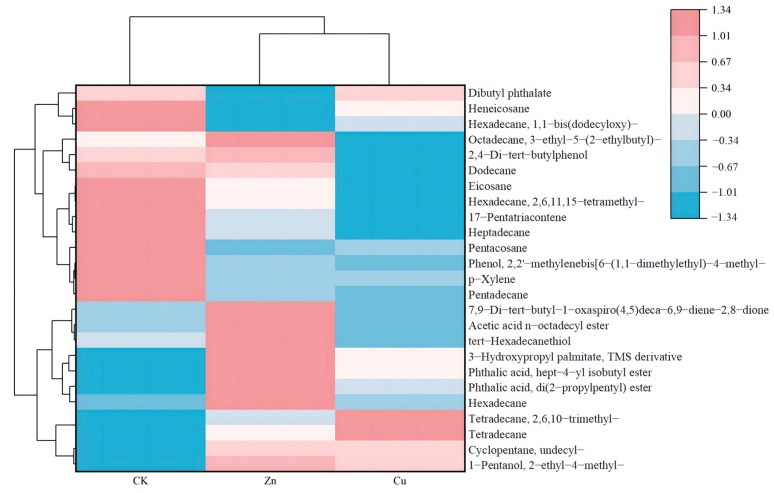
Heat map analysis of the relative amount of root exudates in CK (control), Zn (8 mg/L) and Cu (1 mg/L) treatment for 20 days. CK means the plants cultivated in the Hoagland solution without metal solution exposure.

## Data Availability

Data will be made available on request.

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
