# Peer review of "Responses in Plant Growth and Root Exudates of Pistia stratiotes under Zn and Cu Stress"

_plants, 2024, doi:10.3390/plants13050736_

Round 1
Reviewer 1 Report
Comments and Suggestions for Authors
Author Response
Responses to Reviewers:
Reviewer 1:
Comments and suggestion to authors:
for the manuscript (ID plants-2823816) “Responses in Plant Growth and Root Exudates of Pistia stratiotes under Zn and Cu Stress” written by Yujie Wang , Pan Zhang , Canhua Yang , Yibai Guo , Panpan Gao , Tong Wang , Yu Liu , Lina Xu , Gongke Zhou.
MS presents the data on the copper and zinc effects on growth, leaf chlorophyll content and fluorescence induction, and root exudate composition of Pistia stratioides which is considered as a threat to biodiversity in invaded habitats and at the same time as a potential in the phytoremediation of heavy metal-polluted wastewater. MS can’t be recommended for publication because requires serius revision.
Thanks so much for your comments and suggestions. We carefully revised the manuscript based on the comments of the Editor and reviewers, and all the changes were marked with blue color in the revised manuscript. After revision, the quality and clarity of the manuscript are clearly improved.
Comments
Raw fluorescence OJIP transients do not include calculated parameters (Fv/Fm and so on) (lines 129, 121), they probably were transferred to a spreadsheet along with them.
Thanks for your suggestion. As suggested, we have amended the description of this experimental method. Please also see Lines 128-132, Page 3 and below: “Raw fluorescence OJIP transients were transferred to a spreadsheet using the Pocket PEA program, including minimal fluorescence (Fo), maximal fluorescence (Fm), performance index based on absorbed light energy (PIabs) and so on. PS II maximum light energy con-version efficiency was calculated as Fv/Fm= (Fm-Fo)/Fm. Fv is a variable fluorescence calculated as Fv = Fm-Fo.”.
The authors write that absorbance was measured at 665 nm for Chl a, 649 nm – for Chl b, and 470 nm – for carotenoids (lines 126, 127). That is not correct. All three absorbances are used in calculation of all three pigment groups.
Thanks for your suggestion. We revised the description of the absorbance detection method. Pleased see Lines 137-138, Page 3, and below: “The absorbance of extract solutions was read at 470, 649 and 665 nm against the solvent (96% ethanol) blank with a.....”.
The results of the effect of Zn and Cu should be presented in the same manner, while the author states values of Fv/Fm after Zn treatment and percentages after Cu treatment, which is not directly comparable and not convenient for perception.(lines 183–188)
Thanks for your suggestion. As suggested, we reviewed the results of the values of Fv/Fm after Zn or Cu treatment in Lines 198-203, Page 5. Pleased see below: “Zn exposure (1, 2, 4, and 8 mg/L) induced a decrease of Fv/Fm and reached the value of 0.75, 0.69, 0.64, 0.57 on the 20th day, respectively. They decreased by 3.85%, 11.54%, 17.95%, and 26.92%, respectively, when compared with the blank control. The treatment concentrations of Cu (0.2, 0.4, 0.8 and 1.0 mg/L) showed also inhibition of the Fv/Fm. After Cu treatment for 20 days, the Fv/Fm values were 0.74, 0.73, 0.72, 0.71, respectively, which were decreased by 5.13%, 6.41%, 7.69%, and 8.97%, respectively, ......”.
Authors write “Chlorophyll b is a light-collecting pigment, whereas chlorophyll a is an antenna pigment, most of which is involved in the transmission of light energy absorption, and a small portion of which is a central pigment involved in the light reaction, thus making the level of its content more important for photosynthesis [34].”(lines 209–213). This sentence is superficial, chlorophyll a can not be characterized as an antenna pigment, it is rather a major antenna pigment, and forms a special pair in a reaction center, which plays a key role in the primary photochemical reactions.
Thanks for your suggestion. As suggested, we added deeper discussion about the Chlorophyll in Lines 231-237, Page 7. Pleased see below: “Chlorophyll a content was noted to be more sensitive to stress, showing highest decrease than chlorophyll b and carotenoids. Similar results have also been reported by Sheetal et al. (2016) [36]. Chlorophyll b is a light-collecting pigment, whereas chlorophyll a is rather a major antenna pigment, and forms a special pair in a reaction center, which plays a key role in the primary photochemical reactions. It is reported that Cu inhibits the synthesis of d-aminolevulinic acid and protochlorophyll reductase resulting in the decrease of chlorophyll a content [37].”
HMs should be stated in ionic form rather than just metals. It is more correct to say that Zn2+ affected photosynthesis, than Zn. In addition, we do not know in what form Cu was used, Cu2+ or Cu+.
Thanks for your suggestion. We understand the concern of the reviewer. We indicated the ionic form of Zn2+ and Cu2+ in Line 98, Page 2. Please see and below: “Zn2+ in the form of ZnSO4·7H2O and Cu2+ in the form of CuSO4·5H2O”. And HMs “Zn” and “Cu” used in the manuscript were supported by the similar literature.
Shentu, J., Li, X., Han, R., Chen, Q., Shen, D., Qi, S. Effect of site hydrological conditions and soil aggregate sizes on the stabilization of heavy metals (Cu, Ni, Pb, Zn) by biochar. Science of the Total Environment 2022, 802, 149949.
Liu, Y., Liu, D., Zhang, W., Chen, X., Zhao, Q., Chen, X., Zou, C. Health risk assessment of heavy metals (Zn, Cu, Cd, Pb, As and Cr) in wheat grain receiving repeated Zn fertilizers. Environmental Pollution 2020, 257, 113581.
Kim, T., Kim, T., Zoh, K. Removal mechanism of heavy metal (Cu, Ni, Zn, and Cr) in the presence of cyanide during electrocoagulation using Fe and Al electrodes. Journal of Water Process Engineering 2020, 33, 101109.
MS needs extensive grammatical, semantic, and stylistic editing
Thank for the constructive comments and valuable suggestions! As suggested, we re-checked the language in this manuscript.

Reviewer 2 Report
Comments and Suggestions for Authors
Dear Authors,
Interesting research, but basic, conclusions too laconic. Pistia is also known for its ability to remediate pollutants, including metals, among others. Cu., perhaps the research should be supplemented with this context. The proper complement to chemical analyzes is the measurement of chlorophyll, because it translates into the assessment of the condition of plants and their response to stress caused by pollution (apart from accumulation and visible morphological changes). Best regards
Author Response
Responses to Reviewers:
Reviewer 2:
Interesting research, but basic, conclusions too laconic.
Thanks for your affirmation and suggestion. As suggested, we added the deeper conclusion in Lines 317-320, Page 9. Please see and below: “In this study, the concentrations of Zn and Cu that induced the negative influence on P. stratiotes is significantly higher than that in the environment. These results will be useful for better understanding the accumulation-effect of Zn and Cu in P. stratiotes under long time exposure in the nature environment.”.
Pistia is also known for its ability to remediate pollutants, including metals, among others. Cu., perhaps the research should be supplemented with this context.
Thanks for your suggestion. As suggested, we added more information about the remediation of P. stratiotes on pollutants in lines 74-80, Page 2. Please see and below: “P. stratiotes are tolerant to various environmental conditions and can accumulate or absorb the contaminants from the water environment through their whole body, which is regard-ed as the low-cost and solar energy-driven phytoremediation[20]. It is reported that P. stratiotes presented differential accumulation and tolerance levels under different metals (Ag, Cd, Cr, Cu, Hg, Ni, Pb and Zn) treatment conditions [21]. The removal rates of Cr and Cu by the absorption of P. stratiotes were 77.3%, 80.9%, respectively [22].”.
The proper complement to chemical analyzes is the measurement of chlorophyll, because it translates into the assessment of the condition of plants and their response to stress caused by pollution (apart from accumulation and visible morphological changes). Best regards
Thanks for your positive and valuable comments.

Reviewer 3 Report
Comments and Suggestions for Authors
Although the authors have made an effort in writing the study, it has serious and insurmountable shortcomings, as there is no coherence between the poorly specified experimental design and the results. For these reasons it is my opinion that the manuscript should be rejected.
Author Response
Responses to Reviewers:
Reviewer 3:
Although the authors have made an effort in writing the study, it has serious and insurmountable shortcomings, as there is no coherence between the poorly specified experimental design and the results. For these reasons it is my opinion that the manuscript should be rejected.
Thanks so much for your comments and suggestions. We carefully revised the manuscript based on the comments of the Editor and reviewers. For instance, we added the plant analysis information in Lines 95-96, Page 2, and Lines 105-106, Page 3. Please see below: “The plant material was analyzed at the beginning of the experiment.”; “At the different treated time, the plants in the different treatment groups were collected and further analyzed.”. we clearly defined the control groups in the experiments. Please see Lines 100-101, Page 3, and below: “The healthy plants cultivated in the Hoagland solution without metal solutions exposure were marked as control group (CK).”. We added deeper discussion about the Chlorophyll in Lines 231-237, Page 7. Pleased see below: “Chlorophyll a content was noted to be more sensitive to stress, showing highest decrease than chlorophyll b and carotenoids. Similar results have also been reported by Sheetal et al. (2016) [36]. Chlorophyll b is a light-collecting pigment, whereas chlorophyll a is rather a major antenna pigment, and forms a special pair in a reaction center, which plays a key role in the primary photochemical reactions. It is reported that Cu inhibits the synthesis of d-aminolevulinic acid and protochlorophyll reductase resulting in the decrease of chlorophyll a content [37].” We added the deeper conclusion in Lines 317-320, Page 9. Please see and below: “In this study, the concentrations of Zn and Cu that induced the negative influence on P. stratiotes is significantly higher than that in the environment. These results will be useful for better understanding the accumulation-effect of Zn and Cu in P. stratiotes under long time exposure in the nature environment.”. In addition, we also re-checked the language in this manuscript. After revision, the quality and clarity of the manuscript are clearly improved.

Round 2
Reviewer 1 Report
Comments and Suggestions for Authors
In general, the responses are accepted.
The only suggestion is, in the sentence
“Raw fluorescence OJIP transients were transferred to a spreadsheet using the Pocket PEA program, including minimal fluorescence (Fo), maximal fluorescence (Fm), performance index based on absorbed light energy (PIabs) and so on.",
the word "including" should be replaced with the phrase "along with" or similar, because, as I noted, raw OJIP transients do not include PIabs.
English proofreading is required to avoid similar mistakes, which are insignificant yet complicated reading.

English proofreading is required
Author Response
Thanks very much for your positive and valuable comments. As suggested, we replaced the “including” with “along with” in this sentence. In addition, the manuscript was proofread by two colleagues fluent in English writing. The quality and clarity of the manuscript are clearly improved. Thanks again!

Reviewer 3 Report
Comments and Suggestions for Authors
I find that the manuscript still has the same weaknesses as the original manuscript even though the authors have made a minimal attempt to correct it. However, they have not corrected the fundamentals, such as the experimental design.
Unfortunately, my opinion is that the manuscript does not meet the quality required for an article in Plants.
Author Response
Thanks for your suggestion. We understand the concern of the reviewer. And We carefully revised the manuscript based on the comments of the Editor and reviewers. In this work, we added the detailed information of experimental design, and added more more analysis and discussion to make clearer and stronger statements in the “Results and Discussion” section. We made carefully revisions to improve the manuscript. After revision, the quality and clarity of the manuscript are clearly improved.
